# Adaptive Policy Backbone via Shared Network

**Bumgeun Park**[1]   **Donghwan Lee**[1]

## Abstract

Reinforcement learning (RL) has achieved impressive results across various domains, yet the resulting policies often fail to generalize beyond the specific tasks encountered during training. This lack of robustness limits their deployment in real-world scenarios where diverse and unpredictable task demands exist. In this work, we provide a theoretical analysis of policy networks under Markov Decision Processes (MDPs) and demonstrate that adapting only the linear layers placed before and after a policy backbone is sufficient for task adaptation. Based on this insight, we propose the Adaptive Policy Backbone (APB), which consists of a frozen backbone paired with lightweight, task-specific pre- and post-backbone linear layers. Our results demonstrate that learning only these lightweight task-specific linear layers is sufficient to achieve performance on par with standard RL, even when the backbone is randomly initialized. Furthermore, we find that this structural constraint can enhance the generalization capability of the resulting policies. This advantage extends to out-of-distribution tasks, where representative meta-RL baselines often struggle.

## 1. Introduction

Reinforcement learning (RL) has achieved remarkable success in acquiring complex behaviors across various domains, ranging from games (Mnih et al., 2015; Vinyals et al., 2019; OpenAI et al., 2019b) to robotic control (Levine et al., 2016; OpenAI et al., 2019a). Despite these advancements, a fundamental challenge remains: policies trained via standard RL often fail to generalize when they encounter the specific tasks that lie outside the narrow distribution experienced during training. This lack of robustness creates a significant barrier to practical deployment, where real-world systems must handle unpredictable task horizons and varying environmental conditions that frequently deviate from the training setup.

To enhance this generalization capability, a common strategy is to leverage *priors*, such as pre-collected datasets (Levine et al., 2020) or reference policies (Xie et al., 2021; Kalashnikov et al., 2018), which encapsulate a broader understanding of the environment. In practice, however, the deployment task often differs from that represented by the dataset or reference policy; such *task mismatch* can substantially diminish the utility of these priors. To leverage priors despite this mismatch, several approaches have been proposed in the context of meta-RL (Wang et al., 2017; Duan et al., 2016; Finn et al., 2017; Rakelly et al., 2019), which aim to leverage prior knowledge for efficient adaptation, thereby generalizing behavior outside the training distribution.

Despite progress on handling task mismatch, many methods still struggle with adapting to out-of-distribution (OOD) tasks and thus fail to leverage prior knowledge, because they implicitly assume that training and deployment tasks are drawn from the same distribution, which is rarely realistic. For example, Finn et al. (2017) evaluates policy adaptation on HalfCheetah-vel, where training tasks are reaching target velocities uniformly sampled from $[0, 3]$ and, after training, the policy is adapted to new targets drawn from the same range. However, one typically desires methods that adapt effectively even when tasked with moving backward (negative target velocity), which is fundamentally OOD. This gap highlights the need for methods that reliably adapt to tasks beyond those encountered during training.

In this paper, we propose the **A**daptive **P**olicy **B**ackbone (APB), a transferable policy backbone that employs a meta-initialization to enhance the generalization capability of policies, even on OOD tasks. As illustrated in Figure 1, APB consists of a backbone shared across meta-training tasks, together with task-specific linear layers placed before and after it. This structure is commonly leveraged in multi-task learning to facilitate inductive transfer (Caruana, 1997), where knowledge gained from one task helps improve the learning of others through a shared representation (Baxter, 2000). However, while existing literature primarily focuses on how this parameter sharing benefits in-

---

[1]School of Electrical Engineering, Korea Advanced Institute of Science and Technology (KAIST), Daejeon, Republic of Korea. Correspondence to: Donghwan Lee <donghwan@kaist.ac.kr>.

*Proceedings of the 43rd International Conference on Machine Learning*, Seoul, South Korea. PMLR 306, 2026. Copyright 2026 by the author(s).

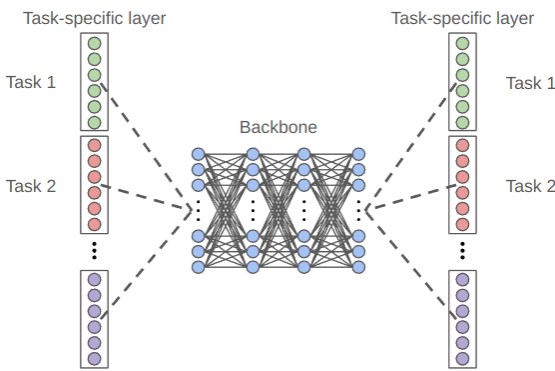

Task-specific layer

Task 1

Task 2

Backbone

Task-specific layer

Task 1

Task 2

*Figure 1.* APB architecture: a shared backbone with task-specific linear layers.

distribution multi-task performance, a rigorous theoretical analysis of its impact on OOD generalization—especially when adapting only task-specific layers while preserving a fixed backbone—remains largely under-explored. In this work, we show that effective task adaptation can be achieved by keeping the shared backbone fixed and adapting only task-specific pre- and post-backbone linear layers.

The main contributions of this paper are as follows:

- **Structural constraints and theoretical analysis:** We demonstrate that training only the task-specific pre- and post-backbone linear layers while preserving the backbone parameters is sufficient for effective task adaptation. This approach aligns with recent advancements in parameter-efficient fine-tuning (PEFT), which have shown that updating a minimal subset of parameters can be effective while preserving pre-trained knowledge. We provide a theoretical analysis in a simplified setting alongside empirical results in complex environments. Although the theoretical analysis does not directly extend to the full complexity of deep RL, it offers valuable intuition regarding why such a modular architecture can facilitate robust transfer. Notably, we find that these structural constraints are sufficiently strong that even a randomly initialized backbone can yield competitive policies, highlighting the important role of structural constraints in policy adaptation.

- **Enhancing generalization capability:** We show that APB improves the generalization capability of policies compared to standard RL, particularly on OOD tasks. This is achieved by leveraging a meta-initialized backbone that captures transferable environmental features and freezing it during adaptation to preserve these representations. We validate this by evaluating APB in extended environments where episode lengths significantly exceed those used in training, forcing the policy to operate in previously unencountered states. Further-

more, we demonstrate APB's robustness through a behavior cloning (BC) evaluation with narrowly concentrated OOD expert demonstrations—a regime where vanilla BC often struggles to generalize.

- **Adaptation to OOD tasks:** We show that APB can adapt to OOD tasks where representative meta-RL baselines often struggle. While conventional meta-RL baselines degrade when the deployment task deviates from the meta-training distribution, APB remains effective under these shifts.

## 2. Related Works

**Parameter-Efficient Fine-Tuning.** Fine-tuning is a widely used approach to adapt pre-trained models to downstream tasks by adjusting model parameters. However, full fine-tuning can be inefficient and often leads to catastrophic forgetting, where previously learned knowledge is overwritten and performance deteriorates. Recently, parameter-efficient fine-tuning (PEFT) has emerged as a promising alternative. PEFT enables adaptation to downstream tasks while mitigating catastrophic forgetting, updating only a small subset of parameters and freezing the rest. Previously, PEFT has been implemented by making only adapter parameters learnable (Zhang et al., 2024; Hu et al., 2022; Sung et al., 2022; Houlsby et al., 2019), or by updating only the bias terms (Zaken et al., 2022), a single layer (Lee et al., 2023; Kumar et al., 2022), or a layer block (Peng et al., 2024). Due to its effectiveness on downstream tasks, PEFT is utilized in various areas such as LLM (Fu et al., 2023; Zhang et al., 2024; Hu et al., 2022; Houlsby et al., 2019; Zaken et al., 2022), vision (Lee et al., 2023; Sung et al., 2022; Kumar et al., 2022; Peng et al., 2024; Jia et al., 2022; Jie & Deng, 2023; Wang et al., 2023), robot learning for sim2real (Truong et al., 2021; Sharma et al., 2023), and meta-RL (Zintgraf et al., 2019; Raghu et al., 2020).

**Transfer Learning.** Transfer learning (TL) is an approach for transferring knowledge learned in a source domain to accelerate learning in a target domain. Because TL aims to transfer informative representations that improve generalization, these ideas have been applied to meta-RL (Devin et al., 2017; Borsa et al., 2019; Zintgraf et al., 2019; Raghu et al., 2020). Devin et al. (2017) pre-train modular policy components across tasks and dynamically reassemble them for new tasks. Zintgraf et al. (2019); Raghu et al. (2020) pre-train policies and adapt by fine-tuning only a subset of parameters to enable fast adaptation. Borsa et al. (2019) leverage successor features to promote generalization across tasks. However, successor-feature methods can degrade under drastic changes in the reward function across domains (Lehnert et al., 2017; Borsa et al., 2019). Distinct from these works, we provide a formal theoretical analysis

of the policy structure to justify its transferability. While previous methods primarily focus on empirical success or representation-level transfer, we investigate the matrix-level structure of policies to demonstrate why a fixed backbone paired with linear transformations is sufficient for cross-task adaptation. This theoretical grounding allows APB to achieve reliable adaptation even when reward structures deviate significantly from the training setup.

**Meta Learning.** Meta-learning aims to train models across a variety of tasks to enable rapid adaptation to new, unseen tasks, ranging from learning effective parameter initializations (Finn et al., 2017; Ravi & Larochelle, 2017) to discovering task-specific update rules (Mishra et al., 2018). In the context of RL, meta-RL has evolved from recurrent architectures that encode task dynamics (Wang et al., 2017; Duan et al., 2016) to gradient-based methods like Model-Agnostic Meta-Learning (MAML) (Finn et al., 2017), which seeks an optimal initialization for fast adaptation. More recent approaches utilize context encoders for posterior inference (Rakelly et al., 2019; Zintgraf et al., 2020) or Transformer-based architectures for in-context learning (Wang et al., 2024; Xu et al., 2022). Despite these advancements, most meta-RL methods frequently struggle with distribution shifts because their adaptation process—which often involves updating the entire network or relying on specific context distributions—can distort the learned priors when encountering OOD tasks. APB addresses this challenge through its unique structural design, which allows the policy to effectively leverage a meta-trained initialization even under significant distribution shifts. By adjusting only the pre- and post-backbone linear layers while keeping the backbone frozen, APB ensures that the invariant environmental representations captured during meta-training are preserved and utilized without distortion.

# 3. Preliminaries

## 3.1. Problem Formulation

We consider a Markov decision process (MDP) defined by the tuple $(\mathcal{S}, \mathcal{A}, P, r, \gamma)$. $\mathcal{S}$ is the state space, $\mathcal{A}$ is the action space, $P(\cdot|s,a)$ is the transition kernel over $\mathcal{S}$, $r : \mathcal{S} \times \mathcal{A} \to [r_{\min}, r_{\max}]$ is the reward function, and $\gamma \in (0,1)$ is a discount factor that controls the weighting of future rewards. At each time step $t$, given state $s_t \in \mathcal{S}$, the agent chooses an action $a_t \in \mathcal{A}$ according to a (possibly stochastic) policy $\pi(a|s)$. The state then transitions from $s_t$ to $s_{t+1}$ according to $P(s_{t+1}|s_t, a_t)$, and the agent receives a reward $r(s_t, a_t, s_{t+1})$. For brevity, we also use $s'$ to denote the next state $s_{t+1}$. Moreover, the state-value function under policy $\pi$ is defined as

$$v^\pi(s_t) = \mathbb{E}_{\substack{a_k \sim \pi(\cdot|s_k) \\ s_{k+1} \sim P(\cdot|s_k, a_k)}} \left[ \sum_{k=t}^{\infty} \gamma^{k-t} r(s_k, a_k, s_{k+1}) \,\Big|\, s_t \right]$$

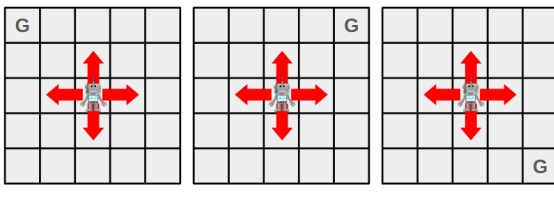

*Figure 2.* Toy example demonstrating MDPs that are isomorphic under state permutation.

The objective of RL is to learn a policy $\pi$ that maximizes the expected return $\mathbb{E}_{s_0 \sim \rho_0}[v^\pi(s_0)]$, where $\rho_0$ denotes the initial-state distribution.

Here, we formalize the meta-RL setting as follows. Let $p(\mathcal{T})$ denote a distribution over tasks, where each task $\mathcal{T}_i \sim p(\mathcal{T})$ corresponds to an MDP $(\mathcal{S}, \mathcal{A}, P, r_\mathcal{T}, \gamma)$. In this work, to isolate the structural mechanisms required for adapting to new objectives, we consider task variations arising primarily from differences in reward functions. This corresponds to a practically relevant setting in which tasks share the same environment dynamics but differ in the agent's goal or objective. During meta-training, we sample $N$ tasks $\{\mathcal{T}_i\}_{i=1}^N \sim p(\mathcal{T})$, and interact with these tasks to learn shared parameters. At evaluation, a novel task $\mathcal{T}_o \sim q(\mathcal{T})$ is drawn from a distribution $q(\mathcal{T})$ that differs from $p(\mathcal{T})$ (i.e., OOD with respect to $p(\mathcal{T})$; e.g., supp $(q) \not\subseteq$ supp $(p)$).

## 3.2. Matrix Expression

Although the MDP considered in this paper has continuous spaces, we adopt a discrete setting for matrix-based analysis. Under policy $\pi$, we retain the notation $v^\pi(s)$ for the scalar state-value function and introduce $V^\pi \in \mathbb{R}^{|\mathcal{S}| \times 1}$ as its vectorized representation over the state space. We further let $R^\pi \in \mathbb{R}^{|\mathcal{S}| \times 1}$ and $P^\pi \in \mathbb{R}^{|\mathcal{S}| \times |\mathcal{S}|}$ denote the expected reward vector and the state transition matrix, respectively. Assuming sufficiently long or infinite-horizon episodes, $V^\pi$ can be expressed as:

$$V^\pi = (I_{|\mathcal{S}|} - \gamma P^\pi)^{-1} R^\pi$$

where $I_{|\mathcal{S}|} \in \mathbb{R}^{|\mathcal{S}| \times |\mathcal{S}|}$ is the identity matrix. Following Wang et al. (2007); Luan et al. (2019); Lim & Lee (2024), let $\Pi^\pi \in \mathbb{R}^{|\mathcal{S}| \times |\mathcal{S}||\mathcal{A}|}$ be the policy matrix as follows:

$$\Pi^\pi = \text{diag}\big(\pi(\cdot|s_1)^\top, \cdots, \pi(\cdot|s_{|\mathcal{S}|})^\top\big) \in \mathbb{R}^{|\mathcal{S}| \times |\mathcal{S}||\mathcal{A}|}$$

Let $R \in \mathbb{R}^{|\mathcal{S}||\mathcal{A}| \times 1}$ be the state-action reward vector with $R_{(s,a)} = r(s,a)$. Then

$$P^\pi = \Pi^\pi P, \quad R^\pi = \Pi^\pi R$$

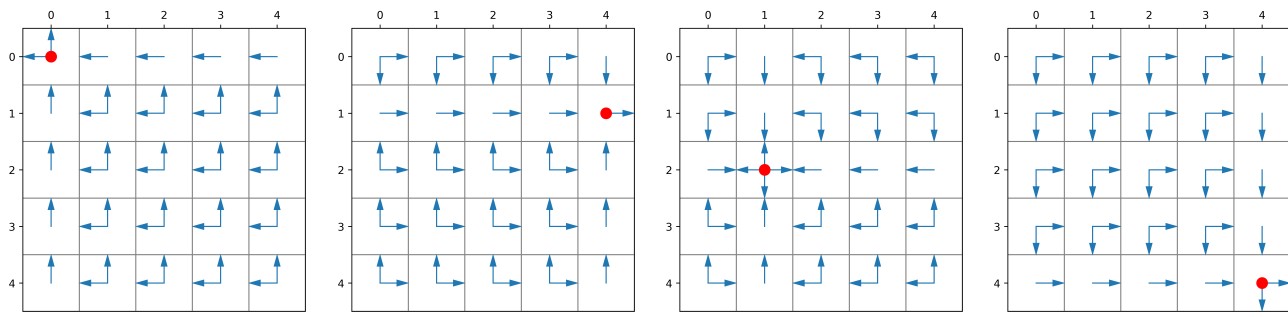

*(a)* Policy plot extracted from matrix $\Pi^{\pi_1}$

*(b)* Policy plot extracted from matrix $A_2\Pi^{\pi_1}B_2$.

*(c)* Policy plot extracted from matrix $A_3\Pi^{\pi_1}B_3$

*(d)* Policy plot extracted from matrix $A_4\Pi^{\pi_1}B_4$

*Figure 3.* Policy plots for different goal positions depicted by a red dot. An agent choose one of the four actions(up, down, left, right) and when face the wall, it stay. $A_i$ and $B_i$ represent matrix introduced in (3)

## 4. Adaptive Policy Backbone

We propose Adaptive Policy Backbone (APB), a simple yet effective transferable policy backbone designed to improve generalization capability over standard RL and to enable adaptation to OOD tasks where existing meta-RL methods often fail. We argue that updating only the first and last layers of the policy network—while freezing the backbone to preserve prior knowledge—suffices for effective adaptation to new tasks. The key intuition is that although each optimal policy is task-specific, policies share a common component induced by structural similarities in the underlying MDPs; this shared structure reduces the need for extensive retraining on new tasks. Structurally, APB is analogous to D'Eramo et al. (2020), which uses a backbone shared across training tasks together with task-specific nonlinear modules placed before and after the backbone. While that line of work emphasizes how shared knowledge aids per-task training, APB replaces the non-linear modules with linear layers for greater parameter efficiency and targets adaptation to OOD tasks in a meta-learning setting. To support our main claim, we provide a theoretical analysis of the policy structure in a simple environment, and extend our study to more complex environments through empirical experiments.

### 4.1. Theoretical study of the policy structure

We begin our analysis with a simple environment with finite state and action spaces. Using the policy-matrix decomposition derived in Section A.1, we obtain the following relation between two policies.

**Lemma 4.1.** *Let* $\Pi^{\pi_1}, \Pi^{\pi_2} \in \mathbb{R}^{|\mathcal{S}| \times |\mathcal{S}||\mathcal{A}|}$ *be the policy matrices for any* $\pi_1$ *and* $\pi_2$, *respectively, and let matrix* $A \in \mathbb{R}^{|\mathcal{S}| \times |\mathcal{S}|}$ *satisfies* $AV^{\pi_1} = V^{\pi_2}$. *Then there exists another matrix* $B \in \mathbb{R}^{|\mathcal{S}||\mathcal{A}| \times |\mathcal{S}||\mathcal{A}|}$ *such that*

$$A\Pi^{\pi_1}B = \Pi^{\pi_2} \qquad (1)$$

Here, $\pi_1$ and $\pi_2$ denote the **source policy** (e.g., an existing

policy learned from source tasks) and the **target policy** (the policy to be adapted), respectively. Lemma 4.1 serves as the key intermediate result for deriving Theorem 4.3. It shows that if the value vectors of two policies are related by a linear transformation $A$, then their policy matrices are also related through linear transformations $A$ and $B$. In this sense, a value-level correspondence induces a policy-level correspondence. To obtain a more interpretable state-wise expression from this matrix relation, we adopt the following restrictive assumption.

**Assumption 4.2.** $A$ is a permutation matrix (i.e., the two MDPs are isomorphic up to a permutation of the state space).

An illustrative toy example is provided in Figure 2.

**Theorem 4.3.** *Under Assumption 4.2, let* $e_s \in \mathbb{R}^{|S|}$ *denote the one-hot vector whose* $s$-th *entry is* 1 *and all other entries are* 0. *Then* $\pi_2$ *can be expressed with any policy* $\pi_1$ *as*

$$\pi_2(\cdot \mid s) = h\Big(\pi_1\big(\cdot \mid g(e_s)\big)\Big)$$

*where* $g : \mathbb{R}^{|S|} \to \mathbb{R}^{|S|}$ *is a linear map (e.g.,* $g(e_s) = e_s^\top A$*) and* $h : \mathbb{R}^{|\mathcal{A}|} \to \mathbb{R}^{|\mathcal{A}|}$ *is a (possibly state-dependent) linear map.*

Theorem 4.3, obtained by combining Lemma 4.1 with Assumption 4.2, provides a structural interpretation of adaptation in this simplified setting. It shows that the target policy can be expressed from the source policy through linear transformations before and after the policy map. This motivates the APB architecture, in which task adaptation is realized through lightweight linear transformations around a shared nonlinear backbone. Based on this theoretical groundwork, we can now formally define the practical implementation of our architecture. Specifically, we parameterize the policy as a composition of linear and nonlinear maps:

$$\pi = f_{\text{linear}}^{\text{out}} \circ f \circ f_{\text{linear}}^{\text{in}}$$

where $f$ denotes a nonlinear backbone, and $f_{\text{linear}}^{\text{in}}$ and $f_{\text{linear}}^{\text{out}}$ denote input and output linear transformations, re-

spectively. Because the composition of linear maps is associative, applying linear transformations before and after $\pi$ can be equivalently viewed as applying linear transformations immediately before and after the nonlinear backbone. We therefore absorb $f_{\text{linear}}^{\text{out}}$ into $h$ and $f_{\text{linear}}^{\text{in}}$ into $g$, and write

$$h \circ \pi \circ g = \left(h \circ f_{\text{linear}}^{\text{out}}\right) \circ f \circ \left(f_{\text{linear}}^{\text{in}} \circ g\right) =: h \circ f \circ g.$$

Here, consistent with Theorem 4.3, $g$ and $h$ correspond to the linear transformations on the input/state representation side and the output/action side, respectively.

Although Assumption 4.2 is restrictive and such isomorphic MDPs are rare in practice, the result offers useful intuition for why freezing the backbone and adapting only linear layers can still be effective. We complement this intuition with extensive empirical results in more practical environments.

### 4.2. Random backbone

A notable implication of Theorem 4.3 is that, in its stated setting, it imposes no specific conditions on the policy $\pi_1$. Interestingly, we observe that even a *randomly initialized* backbone can yield competitive policies when paired with appropriate linear pre- and post-mappings. While Theorem 4.3 is stated under an idealized setting, this empirical finding underscores the role of the architectural constraint in practice. This phenomenon suggests that the structural constraints of the APB architecture—specifically the use of a fixed high-dimensional projection via the backbone followed by task-specific linear adaptation—act as a powerful inductive bias. In this regime, the frozen backbone functions as a set of fixed, non-linear random features. While a meta-initialized backbone provides priors for complex OOD generalization, the unexpected effectiveness of the random backbone highlights that the modularity of the APB structure itself is a primary driver of its adaptability. We demonstrate this "backbone-agnostic" property through extensive experiments in Section 5.2, showing that APB achieves competitive performance compared to standard RL baselines even in the absence of meta-training. While this structural bias is sufficient for adaptation in online RL settings, we further show in Section 5.2 that meta-trained representations become substantially beneficial when generalizing from the restricted data distributions found in behavior cloning.

## 5. Experiments

### 5.1. Experimental Setup

We conduct experiments in the MuJoCo control suite (Todorov et al., 2012) with specific tasks such as reaching goals, or achieving target velocities or directions. These tasks are widely used meta-RL benchmarks adopted in previous studies (Finn et al., 2017; Zintgraf et al., 2019; Raghu

---

**Algorithm 1** Adaptive Policy Backbone (APB)

▷ **Meta-training: Learning shared backbone parameters** $\psi$

Initialize shared backbone parameters $\psi$
Initialize task-specific pre-/post-backbone linear-layer parameters $\rho_i = \{\rho_i^{\text{pre}}, \rho_i^{\text{post}}\}$, critic parameters $\omega_i$, and replay buffer $\mathcal{B}_i$ for each task $\mathcal{T}_i \sim p(\mathcal{T})$
**while** not converged **do**
    **for** each task $\mathcal{T}_i \sim p(\mathcal{T})$ **do**
        Collect transitions $(s, a, r, s')$ in $\mathcal{B}_i$ via policy $\pi(s; \rho_i, \psi)$
        $\omega_i \leftarrow \omega_i - \eta \nabla_{\omega_i} \mathcal{L}_i^{\text{critic}}$ {Update task-specific Q-function}
        $\rho_i \leftarrow \rho_i - \eta \nabla_{\rho_i} \mathcal{L}_i^{\text{actor}}$ {Update linear layers}
    **end for**
    $\psi \leftarrow \psi - \eta \nabla_\psi \left(\frac{1}{n} \sum_{i=1}^n \mathcal{L}_i^{\text{actor}}\right)$ {Update shared backbone}
**end while**

▷ **Online Adaptation: Tuning task-specific pre-/post-backbone linear-layer parameters** $\rho$ **for OOD task** $\mathcal{T}_o$

Initialize pre-/post-backbone linear-layer parameters $\rho = \{\rho^{\text{pre}}, \rho^{\text{post}}\}$ and critic parameters $\omega$ from scratch, and initialize replay buffer $\mathcal{B}$
Load frozen meta-trained shared backbone parameters $\psi$
**for** each step $t = 1, 2, \ldots$ **do**
    Execute $a_t \sim \pi(s_t; \rho, \psi)$ and store $(s_t, a_t, r_t, s_{t+1})$ in $\mathcal{B}$
    Sample batch from $\mathcal{B}$ and compute losses $\mathcal{L}^{\text{critic}}, \mathcal{L}^{\text{actor}}$
    $\omega \leftarrow \omega - \eta \nabla_\omega \mathcal{L}^{\text{critic}}, \rho \leftarrow \rho - \eta \nabla_\rho \mathcal{L}^{\text{actor}}$
**end for**

---

et al., 2020; Song et al., 2020; Rakelly et al., 2019; Zintgraf et al., 2020; Beukman et al., 2023; Xu et al., 2022; Wang et al., 2024), where each task is characterized by a distinct reward function tailored to its specific objective. In contrast to most prior approaches that assess adaptation primarily on in-distribution tasks, our method aims to enhance the generalization capability of policies even under OOD conditions, thereby broadening its practical applicability. To this end, we investigate whether the meta-initialized backbone remains functional in OOD settings by performing adaptation in environments not encountered during meta-training. Additionally, we evaluate this generalization capability by increasing the episode length beyond the training range, thereby forcing the policy to operate in previously unen-

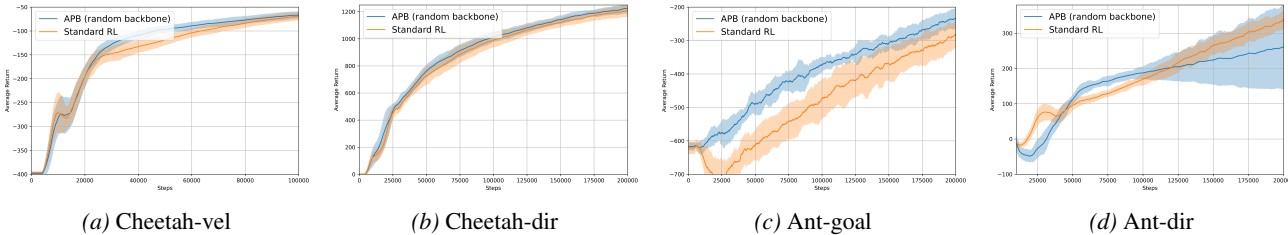

*Figure 4.* Experimental result comparing APB (random backbone) and the standard RL algorithm. Each curve represents the average return over 10 random seeds, with the shaded area indicating one standard deviation from the mean.

countered states. We mainly adopt the code used in Rakelly et al. (2019) with some modifications to implement OOD test protocols. Detailed task specifications, including the OOD settings, are provided in Section B.

We meta-train a shared backbone across meta-training tasks until convergence and, during OOD adaptation, freeze it while adapting task-specific linear layers. We provide a pseudocode for APB in Algorithm 1, in which $\eta$ denotes the learning rate, and both meta-training and online adaptation use the standard SAC critic and actor losses (Haarnoja et al., 2018). For the random-backbone variant, we omit the meta-training stage; the backbone is randomly initialized and then frozen during adaptation. All hyperparameters used in the experiments are listed in Section C.

For the experiments comparing with meta-RL baselines, we evaluate our method against two well-established paradigms: (i) parameterized policy-gradient (PPG) methods that adapt via gradient-based fine-tuning of the parameters (MAML, ANIL, CAVIA; (Finn et al., 2017; Raghu et al., 2020; Zintgraf et al., 2019)) and (ii) black-box methods that adapt by *online inference*—they *continually update* a task-latent from the incoming context while keeping network weights fixed at test time (PEARL, VariBAD; (Rakelly et al., 2019; Zintgraf et al., 2020)). We also include the recent transformer-based Meta-DT (Wang et al., 2024).

The experiments are designed to answer the following questions.

- Is training only the pre- and post-backbone linear layers sufficient for effective task adaptation, even when the backbone parameters are randomly initialized?

- Can APB adapt OOD tasks where existing meta-RL baselines typically fail?

- Can APB improve generalization capability of policy on OOD tasks?

### 5.2. Experimental Results

**Adaptation by training only pre- and post-backbone linear layers** As discussed in Section 4.2, we investigate whether adapting only the pre- and post-backbone

linear layers is sufficient, even with a randomly initialized backbone. In this experiment, the APB backbone is randomly initialized and frozen, while only the pre- and post-backbone linear layers are optimized; we compare this setting against a standard RL baseline of comparable network scale trained end-to-end. As illustrated in Figure 4, this parameter-efficient approach maintains performance on par with standard RL in most environments, such as Cheetah-vel and Cheetah-dir. While a slight performance degradation is observed in Ant-dir, the proposed method unexpectedly outperforms the baseline in Ant-goal. We suspect that such performance gains stem from the structural constraints of the fixed backbone, which may inherently improve generalization by focusing the learning process on a minimal set of task-specific parameters. To evaluate the advantage of adapting both pre- and post-backbone linear layers over probing-only updates, we additionally compare APB against a linear probing baseline (Kumar et al., 2022) that updates only the post-backbone layer while keeping the randomly initialized backbone fixed. We find that adapting both pre- and post-backbone linear layers yields stronger adaptation performance than probing-only updates, supporting the effectiveness of our proposed structural constraint even in the random-backbone setting. Detailed results are provided in Section D. To further examine whether these constraints indeed contribute to better generalization, we conduct behavior cloning (BC) experiments in the following section, demonstrating robust performance even under restricted expert data distributions.

**Adaptation capability of meta-trained backbone on OOD tasks** Existing meta-RL methods have primarily been investigated within in-distribution settings and often struggle when applied to OOD tasks. As illustrated in Figure 5, PPG methods such as MAML, ANIL, and CAVIA show a negligible increase in average return despite fine-tuning their parameters during adaptation, implying a failure to effectively leverage their pre-trained knowledge. Similarly, black-box algorithms including PEARL, VariBAD, and Meta-DT fail to adapt effectively under these conditions. This failure likely stems from the context encoders' inability to process OOD observations; reward variations shift the task objective and drive policies into unseen regions of

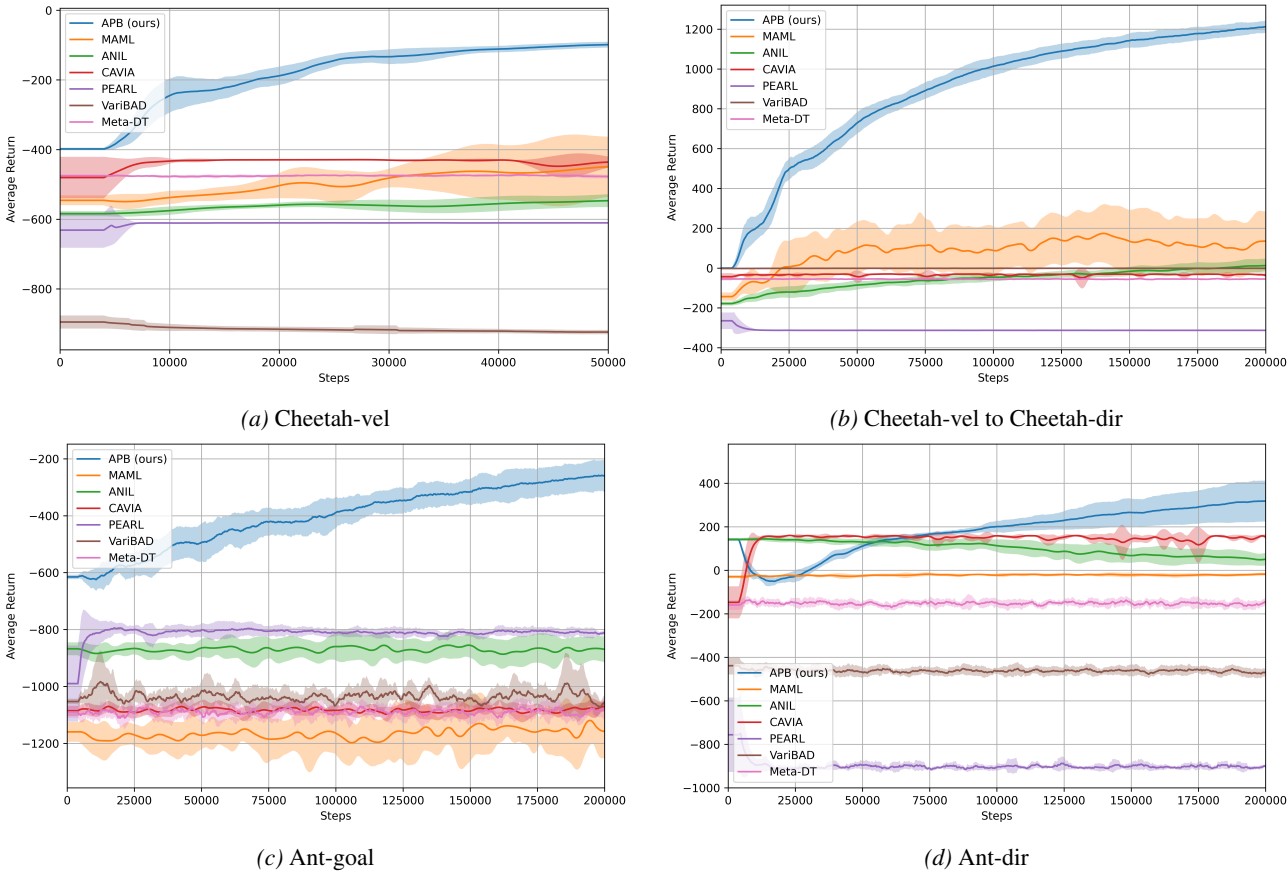

*Figure 5.* Experimental result comparing APB and the existing meta-RL baselines on the out-of-distribution tasks. Each curve represents the average return over 10 random seeds, with the shaded area indicating one standard deviation from the mean.

the state space, yielding data distributions that the encoders were not trained to handle. In contrast, our proposed method demonstrates robust adaptation performance in OOD scenarios. By preserving the meta-trained backbone and only adapting task-specific linear layers, APB successfully utilizes its pre-trained representations to adapt to tasks outside the training distribution without the instability or forgetting seen in existing baselines.

**Improvement of generalization capability** In the previous results, we verified that training only linear layers is sufficient to adapt to new tasks and that a meta-initialized policy backbone can be applied to OOD tasks. Now, we validate whether our approach improves the generalization capability of the policy by extending episode lengths to significantly exceed those used in training ($H_{\text{eval}} > H_{\text{train}}$), forcing extrapolation beyond the support of the training samples and requiring the policy to operate in previously unencountered states. The results are demonstrated in Figure 6. Notably, we found that the randomly initialized backbone exhibits comparable performance. This leads to the question: "How does the meta-trained backbone differ from the randomly initialized one?". To clarify the effect of the meta-trained

backbone, we perform behavior cloning (BC) (Pomerleau, 1988) on (near-)expert demonstrations collected from OOD tasks. Prior work reports that imitation using expert data often fails to generalize because demonstrations cover only a narrow portion of the state–action space (Ross & Bagnell, 2010); accordingly, we deliberately use demonstrations with limited coverage. We compare three configurations: (i) a frozen meta-trained backbone with updated task-specific linear layers, (ii) a frozen randomly initialized backbone, and (iii) a randomly initialized model with all parameters unfrozen. As shown in Figure 7, the meta-pretrained backbone (i) improves generalization over the baselines (ii and iii) on most tasks. While (i) and (ii) show comparable performance in RL settings, their divergence in BC performance suggests that the effectiveness of APB depends on how task-specific linear layers are optimized to leverage the backbone's features. We hypothesize that this gap is mainly due to the difference between interactive data collection and fixed-data generalization: online RL can actively gather task-relevant transitions during adaptation, whereas behavior cloning must adapt from a fixed dataset with limited coverage. As a result, stronger priors in the backbone

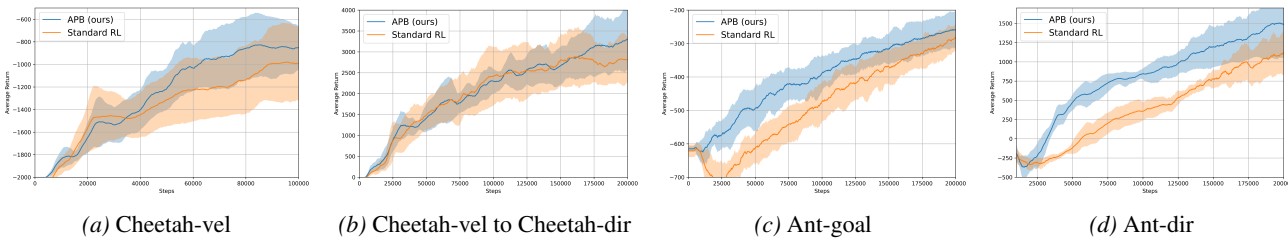

*(a)* Cheetah-vel     *(b)* Cheetah-vel to Cheetah-dir     *(c)* Ant-goal     *(d)* Ant-dir

*Figure 6.* Experimental result comparing APB and the standard RL algorithm on the out-of-distribution tasks. Each curve represents the average return over 10 random seeds, with the shaded area indicating one standard deviation from the mean.

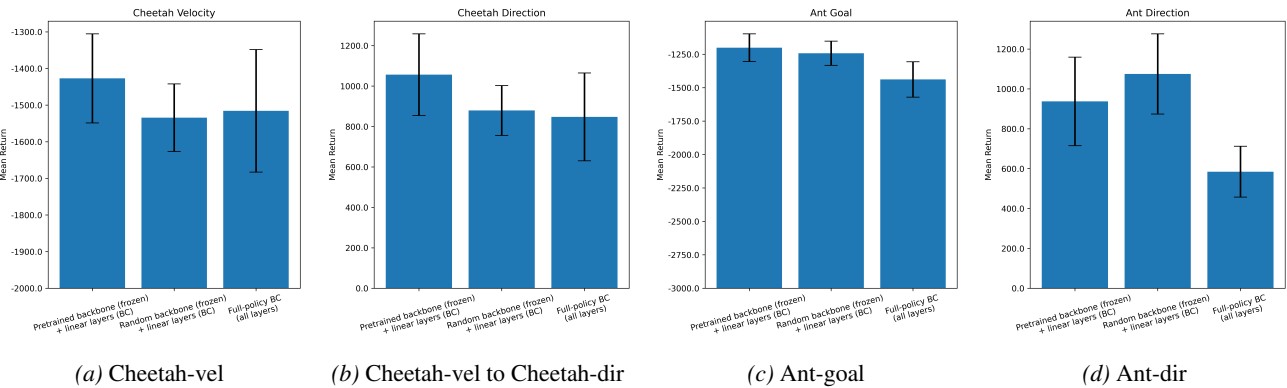

*(a)* Cheetah-vel     *(b)* Cheetah-vel to Cheetah-dir     *(c)* Ant-goal     *(d)* Ant-dir

*Figure 7.* Experimental results on BC. Mean episodic return with 95% confidence intervals across 10 random seeds.

can have a larger impact in BC. Details of the experimental setup are provided in Section E.

## 6. Discussion

In this paper, we propose APB, a transferable policy backbone for improving generalization capability. Our main claims are twofold: (i) updating only linear layers placed before and after a backbone suffices to achieve competitive performance on new tasks, and (ii) APB is capable of generalizing behavior on OOD tasks. We support these claims with a simple theoretical analysis and empirical results on widely used meta-RL benchmarks. Our findings demonstrate that the adaptability of APB stems from a unique synergy between its structural design and meta-learned representations. As discussed in Section 4.2, the structural constraints of APB allow even randomly initialized backbones to function as effective non-linear features for task adaptation. This property highlights that the modularity of our architecture itself serves as a primary driver of its performance. By decoupling the task-specific linear adaptation from the frozen backbone, APB inherently enhances the generalization capability of policies, ensuring robust performance even under significant distribution shifts where existing meta-RL methods typically struggle.

**Limitations.** While APB shows promising results, several limitations remain to be addressed. First, while our theoretical analysis provides fundamental insights, extend-

ing it to deep RL settings presents non-trivial challenges due to the complex non-linear dynamics involved; thus, we primarily rely on extensive empirical evaluations to validate our framework. Second, this study is limited to state-based observations; extending APB to pixel-based domains remains important future work, since such settings require visual representation learning in addition to task adaptation and differ substantially from the continuous-control regime considered here. Third, although APB is parameter-efficient at adaptation time, this does not automatically translate into improved environment-sample efficiency. Improving sample efficiency—e.g., via better meta-objectives that make tasks more linearly separable, stronger initialization of the task-specific layers, or more stable off-policy adaptation—remains an important direction for future work. Lastly, while the random-backbone results suggest that APB's structural design provides a meaningful inductive bias, we do not systematically disentangle the respective contributions of architecture choice and shared meta-parameter quality to the final adaptation performance. In particular, we do not study in depth how performance depends on the choice of backbone architecture or on the optimality of the learned shared parameters. Evaluating different backbone depths, widths, and architectural families, together with a more systematic analysis of meta-parameter quality, remains important future work, particularly for more complex tasks.

## Acknowledgements

This work was supported in part by the Institute of Information Communications Technology Planning Evaluation (IITP) funded by Korean Government under Grant 2022-0-00469, and in part by the BK21 FOUR(Connected AI Education & Research Program for Industry and Society Innovation, KAIST EE, No. 4120200113769)

## Impact Statement

This paper presents work whose goal is to advance the field of machine learning. There are many potential societal consequences of our work, none of which we feel must be specifically highlighted here.

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

# A. Proof

## A.1. Policy-matrix decomposition

**Lemma A.1.** *With $V^\pi = (I_{|\mathcal{S}|} - \gamma P^\pi)^{-1} R^\pi$, $P^\pi = \Pi^\pi P$, and $R^\pi = \Pi^\pi R$, the policy matrix $\Pi^\pi$ can be expressed as*

$$\Pi^\pi = \frac{V^\pi \left(\gamma P V^\pi + R\right)^\top}{\left\|\gamma P V^\pi + R\right\|_2^2} + N$$

*where $\Pi^\pi, N \in \mathbb{R}^{|\mathcal{S}| \times |\mathcal{S}||\mathcal{A}|}$, $P \in \mathbb{R}^{|\mathcal{S}||\mathcal{A}| \times |\mathcal{S}|}$, $V^\pi, R^\pi \in \mathbb{R}^{|\mathcal{S}|}$, $R \in \mathbb{R}^{|\mathcal{S}||\mathcal{A}|}$, $\| \cdot \|_2^2$ denotes the squared $\ell_2$-norm and $N$ satisfies*

$$N(\gamma P V^\pi + R) = \mathbf{0}_{|\mathcal{S}|}.$$

*This equation implies that the row vectors of $N$ are orthogonal to $\gamma P V^\pi + R$.*

*Proof.* From the definition, we have

$$V^\pi = (I_{|\mathcal{S}|} - \gamma P^\pi)^{-1} R^\pi = (I_{|\mathcal{S}|} - \gamma \Pi^\pi P)^{-1} \Pi^\pi R.$$

Multiplying both sides by $(I_{|\mathcal{S}|} - \gamma \Pi^\pi P)$ yields

$$(I_{|\mathcal{S}|} - \gamma \Pi^\pi P)V^\pi = \Pi^\pi R.$$

Expanding the left-hand side, we obtain

$$V^\pi - \gamma \Pi^\pi P V^\pi = \Pi^\pi R.$$

Rearranging the terms gives

$$\Pi^\pi (R + \gamma P V^\pi) = V^\pi. \tag{2}$$

Equation (2) is a linear system in the unknown matrix $\Pi^\pi$. A particular solution is obtained by projecting $V^\pi$ onto the vector $(R + \gamma P V^\pi)$. Therefore, the general solution can be written as

$$\Pi^\pi = \frac{V^\pi (R + \gamma P V^\pi)^\top}{\|R + \gamma P V^\pi\|_2^2} + N,$$

where $\Pi^\pi, N \in \mathbb{R}^{|\mathcal{S}| \times |\mathcal{S}||\mathcal{A}|}$, $P \in \mathbb{R}^{|\mathcal{S}||\mathcal{A}| \times |\mathcal{S}|}$, $V^\pi, R^\pi \in \mathbb{R}^{|\mathcal{S}|}$, $R \in \mathbb{R}^{|\mathcal{S}||\mathcal{A}|}$, and $N$ is any matrix whose rows are orthogonal to $(R + \gamma P V^\pi)$, i.e.,

$$N(R + \gamma P V^\pi) = 0_{|\mathcal{S}|}.$$

This completes the proof.

$\square$

## A.2. Proof of Lemma 4.1

**Lemma 5.1.** *Let $\Pi^{\pi_1}, \Pi^{\pi_2} \in \mathbb{R}^{|\mathcal{S}| \times |\mathcal{S}||\mathcal{A}|}$ be the policy matrices for any $\pi_1$ and $\pi_2$, respectively, and let matrix $A \in \mathbb{R}^{|\mathcal{S}| \times |\mathcal{S}|}$ satisfies $A V^{\pi_1} = V^{\pi_2}$. Then there exists another matrix $B \in \mathbb{R}^{|\mathcal{S}||\mathcal{A}| \times |\mathcal{S}||\mathcal{A}|}$ such that*

$$A \Pi^{\pi_1} B = \Pi^{\pi_2} \tag{3}$$

*Proof.* We start from the decomposition of $\Pi^{\pi_1}$ and $\Pi^{\pi_2}$ given in Lemma A.1:

$$\Pi^{\pi_1} = \frac{V_1 \left(\gamma P V_1 + R_1\right)^\top}{\|\gamma P V_1 + R_1\|_2^2} + N_1,$$

$$\Pi^{\pi_2} = \frac{V_2 \left(\gamma P V_2 + R_2\right)^\top}{\|\gamma P V_2 + R_2\|_2^2} + N_2$$

where $V_1 := V^{\pi_1}$ and $V_2 := V^{\pi_2}$ denote the state-value vectors of policies $\pi_1$ and $\pi_2$, respectively, $R_1 := R^{\pi_1}$ and $R_2 := R^{\pi_2}$ denote the corresponding expected reward vectors, $P \in \mathbb{R}^{|\mathcal{S}||\mathcal{A}| \times |\mathcal{S}|}$, and $N_1, N_2 \in \mathbb{R}^{|\mathcal{S}| \times |\mathcal{S}||\mathcal{A}|}$ are matrices

whose rows are orthogonal to $(\gamma PV_1 + R_1)$ and $(\gamma PV_2 + R_2)$, respectively. We can find a matrix $A \in \mathbb{R}^{|\mathcal{S}| \times |\mathcal{S}|}$ such that $AV_1 = V_2$, and define

$$B = \frac{(\gamma PV_1 + R_1)(\gamma PV_2 + R_2)^\top}{\|\gamma PV_2 + R_2\|_2^2} \in \mathbb{R}^{|\mathcal{S}||\mathcal{A}| \times |\mathcal{S}||\mathcal{A}|}$$

Then

$$
\begin{aligned}
A\Pi^{\pi_1} B &= A \frac{V_1 (\gamma PV_1 + R_1)^\top}{\|\gamma PV_1 + R_1\|_2^2} B + AN_1 B \\
&= \frac{V_2 (\gamma PV_1 + R_1)^\top}{\|\gamma PV_1 + R_1\|_2^2} B + AN_1 B \\
&= \frac{V_2 (\gamma PV_2 + R_2)^\top}{\|\gamma PV_2 + R_2\|_2^2} + AN_1 \underbrace{\frac{(\gamma PV_1 + R_1)(\gamma PV_2 + R_2)^\top}{\|\gamma PV_2 + R_2\|_2^2}}_{\text{rows orthogonal to } (\gamma PV_2 + R_2)}
\end{aligned}
\tag{4}
$$

For the residual term in Equation (4), right-multiplying by $(\gamma PV_2 + R_2)$ yields

$$AN_1 \frac{(\gamma PV_1 + R_1)(\gamma PV_2 + R_2)^\top}{\|\gamma PV_2 + R_2\|_2^2}(\gamma PV_2 + R_2) = AN_1(\gamma PV_1 + R_1)$$

Since the rows of $N_1$ are orthogonal to $(\gamma PV_1 + R_1)$, we have $N_1(\gamma PV_1 + R_1) = \mathbf{0}_{|\mathcal{S}|}$.

$$= A\mathbf{0}_{|\mathcal{S}|} = \mathbf{0}_{|\mathcal{S}|},$$

Therefore, its rows are orthogonal to $(\gamma PV_2 + R_2)$. Hence, setting

$$N_2 := AN_1 \frac{(\gamma PV_1 + R_1)(\gamma PV_2 + R_2)^\top}{\|\gamma PV_2 + R_2\|_2^2}$$

gives

$$A\Pi^{\pi_1} B = \frac{V_2 (\gamma PV_2 + R_2)^\top}{\|\gamma PV_2 + R_2\|_2^2} + N_2 = \Pi^{\pi_2}.$$

This completes the proof. $\square$

### A.3. Proof of Theorem 4.3

**Theorem 5.3.** *Under Assumption 4.2, let $e_s \in \mathbb{R}^{|\mathcal{S}|}$ denote the one-hot vector whose $s$-th entry is $1$ and all other entries are $0$. Then $\pi_2$ can be expressed with any policy $\pi_1$ as*

$$\pi_2(\cdot \mid s) = h\Big(\pi_1\big(\cdot \mid g(e_s)\big)\Big)$$

*where $g : \mathbb{R}^{|\mathcal{S}|} \to \mathbb{R}^{|\mathcal{S}|}$ is a linear map (e.g., $g(e_s) = e_s^\top A$) and $h : \mathbb{R}^{|\mathcal{A}|} \to \mathbb{R}^{|\mathcal{A}|}$ is a (possibly state-dependent) linear map.*

*Proof.* Let $\tilde{\pi}(s) \in \mathbb{R}^{1 \times |\mathcal{S}||\mathcal{A}|}$ denote the zero-padded action-selection probability vector at state $s$, where only the entries corresponding to state $s$ are nonzero. All other entries are zero. For example, if $\mathcal{S} = \{1, 2\}$ and $\mathcal{A} = \{1, 2\}$, and the input state is 1, then the policy network outputs

$$\tilde{\pi}(1) = [\pi(1|1), \pi(2|1), 0, 0]$$

and for state 2, the output is

$$\tilde{\pi}(2) = [0, 0, \pi(1|2), \pi(2|2)]$$

Let $\mathcal{E}(\cdot)$ be the policy extraction operator that extracts the nonzero action-probability block from $\tilde{\pi}(s)$, so that $\mathcal{E}(\tilde{\pi}(s)) = \pi(\cdot|s)$. In this case, we have

$$\mathcal{E}([\pi(1|1), \pi(2|1), 0, 0]) = [\pi(1|1), \pi(2|1)].$$

Since the state is represented as a one-hot vector, we have:

$$e_s^\top \Pi^\pi = \tilde{\pi}(s)$$

Because $\Pi^{\pi_2} = A\Pi^{\pi_1}B$, we have

$$\begin{aligned}
\pi_2(\cdot|s_i) &= \mathcal{E}\left(\tilde{\pi}_2(s_i)\right) \\
&= \mathcal{E}\left(e_{s_i}^\top \Pi^{\pi_2}\right) \\
&= \mathcal{E}\left(e_{s_i}^\top A\Pi^{\pi_1}B\right)
\end{aligned}$$

Let $s_k \in \mathcal{S}$ be the state such that $e_{s_i}^\top A = e_{s_k}^\top$ (i.e., $A$ maps the row corresponding to $e_{s_i}^\top$ to that of $e_{s_k}^\top$). Then

$$\begin{aligned}
\pi_2(\cdot|s_i) &= \mathcal{E}\left(e_{s_k}^\top \Pi^{\pi_1}B\right) \\
&= \mathcal{E}\left(\tilde{\pi}_1(s_k)B\right)
\end{aligned}$$

Since $\tilde{\pi}_1(s_k)$ has support only on the $k$-th state block, $\tilde{\pi}_1(s_k)B$ depends solely on the $k$-th row block of $B$. Hence, regardless of the specific extraction mechanism, applying $\mathcal{E}$ returns a $|\mathcal{A}|$-dimensional *linear* image of $\pi_1(\cdot \mid s_k)$; that is, there exists a matrix $B' \in \mathbb{R}^{|\mathcal{A}| \times |\mathcal{A}|}$ such that

$$\mathcal{E}\left(\tilde{\pi}_1(s_k)B\right) = \pi_1(\cdot \mid s_k)B'$$

We define $g$ by $g(s_i) = s_k$ (equivalently, $s_i A = s_k$), and define $h$ by $h\left(\pi(\cdot|s)\right) = \pi(\cdot|s)B'$. With these definitions, we can express $\pi_2$ as the composition $h \circ \pi_1 \circ g$, which completes the argument. $\square$

# B. Task description

*Table 1.* Task description

| Task (objective) | Meta train | Meta test |
|---|---|---|
| **Cheetah-vel** (Reaching velocity $(v)$) | $v \sim \mathcal{U}(0.0, 3.0)$ | $v = -2.0$ |
| **Cheetah-vel to Cheetah-dir** (Going target direction) | $v \sim \mathcal{U}(0.0, 3.0)$ | backward |
| **Ant-goal** (Reaching goal position $(x, y)$) | $\theta \sim \mathcal{U}(0, \pi)$ $x, y = 3\cos\theta, 3\sin\theta$ | $\theta = 1.5\pi$ $x, y = 3\cos\theta, 3\sin\theta$ |
| **Ant-dir** (Going target direction $(\theta)$) | $\theta \sim \mathcal{U}(0, \pi)$ | $\theta = 1.5\pi$ |

$\mathcal{U}(a, b)$ denotes the uniform distribution over $[a, b]$.

**Cheetah-vel task.** The agent is required to reach a target velocity $v$ using a cheetah model. During meta-training, target velocities are uniformly sampled from $\mathcal{U}(0.0, 3.0)$, and in meta-test, the agent is evaluated at $v = -2.0$.

**Cheetah-vel to Cheetah-dir task.** This task is designed to evaluate the generalization from the Cheetah-vel to the Cheetah-dir setting. The agent is trained to move in randomly sampled target velocities $v \sim \mathcal{U}(0.0, 3.0)$ during meta-training and is tested in the backward direction during meta-test.

**Ant-goal task.** The ant agent is required to reach a goal position $(x, y)$ located on a semicircle of radius 3. For meta-training, the target angle $\theta$ is sampled uniformly from $[0, \pi]$ and the corresponding goal is set as $x = 3\cos\theta, y = 3\sin\theta$. In meta-test, the agent is evaluated at $\theta = 1.5\pi$, that is, $x = 3\cos\theta, y = 3\sin\theta$.

**Ant-dir task.** In this task, the ant must move in a specified direction $\theta$. During meta-training, the direction is uniformly sampled from $\mathcal{U}(0, \pi)$. In meta-test, the agent is evaluated at a fixed direction $\theta = 1.5\pi$.

*Table 2.* Hyperparameters for adaptation

| Hyperparameter | Value(s) |
| --- | --- |
| Batch size | 512 |
| Buffer size | 200000 |
| Learning rate | 0.0001 (Actor); 0.05 (Critic) |
| Number of meta-train tasks | 30 |

## C. Hyperparameters

## D. Linear Probing Baseline Results

Figure 8 reports the comparison between APB and the linear probing baseline in the random-backbone setting.

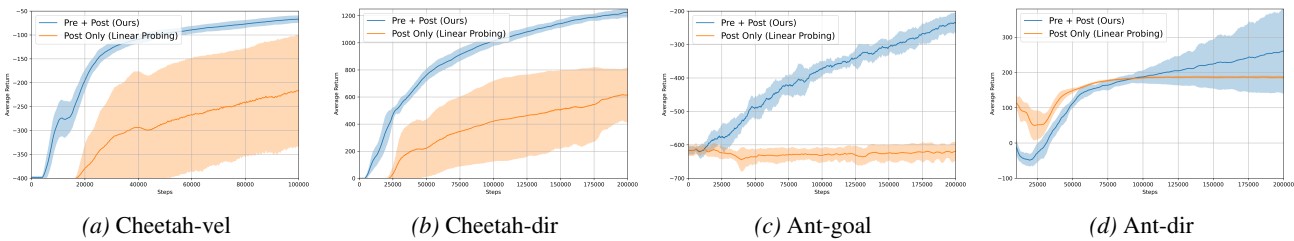

(a) Cheetah-vel        (b) Cheetah-dir        (c) Ant-goal        (d) Ant-dir

*Figure 8.* Experimental result comparing APB and the linear probing baseline in the random-backbone setting. Each curve represents the average return over 10 random seeds, with the shaded area indicating one standard deviation from the mean.

## E. Behavior cloning

**Demonstrations.** We collect (near-)expert demonstrations $\mathcal{D}_t = \{(s, a)\}_{i=1}^N$ for each *OOD* test task by rolling out a (near-)expert policy $\pi_{\exp}$. Each per-task dataset contains $N = 100{,}000$ state–action pairs (transitions) aggregated across multiple episodes. The demonstrations consist of contiguous expert rollouts with episode horizon $H_{\text{demo}}$ and therefore concentrate mass on the expert visitation distribution $d^{\pi_{\exp}}$ (narrow state–action support).

**Policy parameterizations.** We evaluate two models trained on the same demonstrations.

- **APB-BC (ours).** The meta-trained backbone $f_{\text{meta}}$ is *frozen*. We learn only the task-specific linear layers $g$ (pre-backbone) and $h$ (post-backbone).

- **Random APB-BC (ours).** The randomly initialized backbone $f_{\text{random}}$ is *frozen*. We learn only the task-specific linear layers $g$ (pre-backbone) and $h$ (post-backbone). All other hyperparameters and training procedures are identical to APB-BC.

- **Full model (baseline).** Identical architecture but *randomly initialized*; all parameters are unfrozen and updated during training.

**Objective.** Following standard continuous-control BC, we use a deterministic policy and minimize the mean-squared error between actions and expert actions:

$$\mathcal{L}_{\text{BC}}(\theta) \;=\; \frac{1}{|\mathcal{B}|} \sum_{(s,a) \in \mathcal{B}} \left\| \pi_\theta(s) - a \right\|_2^2,$$

where $\mathcal{B}$ is a mini-batch. (For completeness, a stochastic alternative is the negative log-likelihood $\mathcal{L}_{\text{NLL}}(\theta) = -\frac{1}{|\mathcal{B}|} \sum \log \pi_\theta(a \mid s)$, but we do not use it in our experiments.)

**Optimization and preprocessing.** We optimize with Adam (PyTorch), a mini-batch size of 2048, using *task-dependent* learning rates:

$$\text{LR} = \begin{cases} 5 \times 10^{-4}, & \text{Walker-rand-params} \\ 1 \times 10^{-4}, & \text{Hopper-rand-params} \\ 1 \times 10^{-3}, & \text{All other tasks (Cheetah-dir, Cheetah-vel, Ant-goal, Ant-dir).} \end{cases}$$

**Evaluation.** After training, we roll out each policy for episodes with horizon $H_{\text{eval}} > H_{\text{demo}}$ to stress extrapolation beyond the demonstration support (OOD generalization). Performance is measured by episodic return. We report the mean across 10 random seeds and 95% confidence intervals.

## F. Practical implementation of APB

During meta-training, a single policy backbone is shared across all tasks, with task-specific *linear layers* placed before and after the backbone. Each task also has its own Q-functions (critics). After meta-training, we freeze the backbone and train new task-specific linear layers on the meta-test task using the same optimization pipeline; unlike meta-training, only this small subset of policy parameters is updated at adaptation time.

## G. Implementation of meta-RL baselines

We use the authors' public implementations *and* their default hyperparameters for all baselines: MAML (Finn et al., 2017), CAVIA (Zintgraf et al., 2019), PEARL (Rakelly et al., 2019), VariBAD (Zintgraf et al., 2020), and Meta-DT (Wang et al., 2024). ANIL (Raghu et al., 2020) is implemented by modifying our MAML code to freeze the feature backbone during adaptation (inner loop), while keeping the head trainable; we otherwise reuse the same meta-training and adaptation hyperparameters as in MAML. For a fair comparison, we keep the policy network size (hidden depth and units per layer) identical across all methods.

## H. Parameter efficiency

*Table 3.* Comparison of parameter efficiency and computational cost. **Training FLOPs** denotes the estimated floating-point operations per iteration (Forward + Backward). Following standard scaling laws (Kaplan et al., 2020), we estimate backward pass FLOPs as approximately $2\times$ the forward pass FLOPs for the baseline, while APB's cost is scaled down to account for the frozen backbone parameters (i.e., skipping weight gradient computations).

| Environment | Method | Trainable Params | Ratio (%) | Training FLOPs | Time / Iter. |
|---|---|---|---|---|---|
| Cheetah-vel | Baseline | 219,606 | 100.00% | 0.66 M | 5.65 ms |
| | **APB (Ours)** | **6,906** | **3.14%** | **0.45 M** | **4.78 ms** |
| Cheetah-dir | Baseline | 219,606 | 100.00% | 0.66 M | 5.68 ms |
| | **APB (Ours)** | **6,906** | **3.14%** | **0.45 M** | **4.96 ms** |
| Ant-goal | Baseline | 247,708 | 100.00% | 0.75 M | 5.61 ms |
| | **APB (Ours)** | **35,008** | **14.13%** | **0.53 M** | **4.92 ms** |
| Ant-dir | Baseline | 221,908 | 100.00% | 0.67 M | 5.61 ms |
| | **APB (Ours)** | **9,208** | **4.15%** | **0.46 M** | **4.89 ms** |

