# OpenReview forum: "Adaptive Policy Backbone via Shared Network"
_ICML.cc/2026/Conference — ICML 2026 spotlight_

### Official Review · Reviewer_HMfN · 2026-03-09

**Soundness:** 2
**Presentation:** 3
**Significance:** 2
**Originality:** 2
**Overall Recommendation:** 4
**Confidence:** 3

**Summary:**

The maniscript addresses the out-of-distribution generalization problem of standard RL and meta-RL methods. The authors proposed a shared, frozen backbone with task-specific linear layers at the input and output.

however, the key of the proposed method, "fine-tuning few layers of a pre-trained model", is a very common strategy in multi-task learning, meta learning, or adaptation.

**Compliance With Llm Reviewing Policy:**

Affirmed.

**Final Justification:**

The authors' response partly addressed my concern, it's a good piece of work, so i raised my score.

**Key Questions For Authors:**

1. Could the authors provide a more detailed discussion regarding the optimality of meta-parameters? While the manuscript focuses on OOD generalization, the success of the method seems to hinge on the quality of these meta-parameters. It would be beneficial to position meta-parameter optimality as a foundational sub-problem that directly enables the OOD robustness described in the results. For example, "Meta-Reinforcement Learning with Universal Policy Adaptation: Provable Near-Optimality under All-task Optimum Comparator", which is a relevant optimization-based meta-RL method that seeks to identify optimal meta-parameters.

**Limitations:**

see above

**Strengths And Weaknesses:**

**Strengths:**
1. the paper focus on a classic and important problem, the OOD generalisation ability of RL.

**Weaknesses:**
1. The novelty of this approach is limited, as utilizing a frozen backbone with task-specific linear layers is a classic strategy for adaptation and meta-learning. Please refer to a classic work "Surgical Fine-Tuning Improves Adaptation to Distribution Shifts".
2. The selection of baselines in this work appears improper given the current landscape of meta-reinforcement learning (meta-RL). Meta-RL methods are generally categorized into two main paradigms: (i) optimization-based meta-RL and (ii) black-box (or context-based) meta-RL. Optimization-based approaches, such as MAML and its variants, typically involve a nested structure consisting of a policy adaptation algorithm and a meta-algorithm. The meta-algorithm learns a meta-policy that serves as a highly effective initialization for rapid adaptation to new tasks during meta-testing. Conversely, black-box meta-RL aims to train an end-to-end model with fixed parameters that generates task-specific policies based on trajectories collected from new environments.
Given this taxonomy, the authors should reconsider their choice of baselines. At least, PEARL should not be a proper one to be compared with the proposed method.

---

> ### Author Rebuttal · Authors · 2026-03-30
>
> We thank the reviewer for the thoughtful comments. We appreciate the careful engagement with the paper’s positioning and technical scope. Below, we address each point in turn.
>
> **Answer to weakness~1.**
> We agree that the broad idea of combining a frozen/shared backbone with lightweight task-specific adaptation is not entirely new, and we do not intend to claim otherwise. We would also like to note that *Surgical Fine-Tuning Improves Adaptation to Distribution Shifts* is already cited in our paper as relevant prior work [1]. That said, the novelty we claim is more specific. We do not claim that selective fine-tuning itself is new; rather, our contribution is to isolate, in the OOD meta-RL setting, a particularly constrained adaptation structure in which adaptation is restricted to *pre- and post-backbone linear layers* around a frozen shared backbone, and to support this design with both theoretical analysis and empirical validation. In this sense, the contribution of APB is not a new generic adaptation motif, but a more specific structural claim about what form of adaptation can be sufficient for OOD task transfer. We will revise the discussion to clarify more precisely how our contribution differs from prior work such as Surgical Fine-Tuning.
>
> [1] Y. Lee, A. S. Chen, F. Tajwar, A. Kumar, H. Yao, P. Liang, and C. Finn, Surgical Fine-Tuning Improves Adaptation to Distribution Shifts, ICLR 2023.
>
> **Answer to weakness~2.**
> We agree that meta-RL methods are often grouped into optimization-based approaches and black-box/context-based approaches. However, we do not agree that this taxonomy makes PEARL an improper baseline for our study. Our goal is not to compare APB only against methods that adapt through gradient-based parameter updates, but to evaluate it against representative meta-RL paradigms for adaptation to unseen tasks. In particular, black-box/context-based methods such as PEARL and VariBAD do not keep adapting by changing the policy weights at test time, but they nevertheless perform task adaptation by continually updating a task-latent/context variable from trajectories collected in the new environment while keeping the network weights fixed. This is exactly why we included them: although the adaptation mechanism differs from optimization-based meta-RL, they are still representative meta-RL approaches for fast adaptation to unseen tasks. We also note that cross-paradigm comparisons of this kind are standard in the meta-RL literature; for example, prior work such as MetaCURE compares MAML-style optimization-based methods together with PEARL- and VariBAD-style context-based methods under the same benchmark setting [1]. For this reason, we believe PEARL is an intentional and appropriate comparator in our paper, not as an optimization-based baseline, but as a representative context-based meta-RL baseline. We agree, however, that this rationale should have been stated more explicitly, and we will clarify it in the final version.
>
> [1] J. Zhang, J. Wang, H. Hu, T. Chen, Y. Chen, C. Fan, and C. Zhang, MetaCURE: Meta Reinforcement Learning with Empowerment-Driven Exploration, ICML 2021.
>
> **Answer to question~1.**
> We agree that the quality of the shared meta-parameters is important, especially when the backbone is reused across tasks and kept fixed during OOD adaptation. In this sense, learning good shared parameters is certainly a foundational sub-problem for strong adaptation. However, we do not believe that the success of APB can be explained solely by meta-parameter optimality. One of the key findings of our paper is that, in online RL adaptation, even a randomly initialized frozen backbone remains competitive when combined with task-specific linear adaptation. This suggests that the APB structure itself provides a meaningful inductive bias beyond the quality of the learned meta-parameters alone.
>
> At the same time, our results also show that meta-parameter quality matters more strongly in harder fixed-data generalization regimes. In the behavior cloning experiments, the meta-trained backbone shows a clearer advantage over the random one, indicating that stronger shared representations become more important when adaptation must rely on limited-support demonstrations rather than interactive exploration. Our interpretation is therefore that meta-parameter quality and adaptation structure are complementary factors: the former provides transferable shared representations, while the latter determines how robustly those representations are adapted under OOD shift.
>
> We agree that this relationship should be positioned more explicitly. We will therefore revise the discussion to clarify that our work addresses the structural side of OOD adaptation, while optimization-based approaches that study meta-parameter optimality are highly relevant and complementary to our setting.

---

> > ### Author Rebuttal · Reviewer_HMfN · 2026-04-03
> >
> > Thanks for the author's response. Most of my concerns are addressed, it's a good piece of work, I have adjusted my score accordingly.

---

> > > ### Author Response · Authors · 2026-04-03
> > >
> > > Thanks for raising the score! We will incorporate all the discussed points in the revised version. We really appreciate the reviewer's time and effort in helping us strengthen this work.

---

### Official Review · Reviewer_C71Z · 2026-03-10

**Soundness:** 2
**Presentation:** 1
**Significance:** 3
**Originality:** 3
**Overall Recommendation:** 4
**Confidence:** 4

**Summary:**

The paper introduces APB, a reinforcement learning architecture that separates a shared backbone network from small task-specific linear layers. The backbone learns general representations across tasks, while only the lightweight linear head is trained for each new task. This design allows policies to adapt quickly with far fewer parameters while improving generalization to new environments. Overall, the work shows that strong shared representations plus simple task-specific adaptation can outperform fully retrained policies.

**Compliance With Llm Reviewing Policy:**

Affirmed.

**Final Justification:**

The rebuttal has partially addressed my concerns, so I have raised my scores.

**Key Questions For Authors:**

- What are the OOD environments and how are they OOD from the training set?
- Can you explain how algorithms 1 works in detail? What are the head/tail? What are the task-specific Q-function? Are Q-functions entirely task-specific or are they also initialized with the APB backbone? What are the losses you use to update the linear layers, and what is the loss for the actor shared backbone?
- In Figure 4 with the randomly initialized backbone, are you comparing the same architecture and same method with the backbone frozen vs unfrozen? I don't understand how a frozen & randomly initialized feature is better than a learned one (where all parameters are fine-tuned).

**Limitations:**

yes

**Strengths And Weaknesses:**

strengths:
- The PEFT technique discussed in the paper is very interesting and of importance to the community: it is very important to fine-tune efficiently and able to handle OOD environments.
- The paper compares against a valid set of baselines and show strong results
- As far as I know, the method of fine-tuning pre- and post- backbone layers is original.

Weaknesses:
- Presentation: I found the presentation of the method and experimental setting quite unclear and hard to understand. Specifically, I do not understand how the algorithm actually work. Algorithm 1 is referred to in the experiment section, but never explained. Quantities and equations in the algorithm box are never defined. It is also unclear whether you do the "meta-training" stage if the APB backbone is randomly initialized. I would suggest adding a "method section" that explains in detail how the algorithm works, and set it up in the experiment section as well, especially for the distinction on the meta-trained backbone and the randomly initialized backbone. See my questions on this.
- PEFT baselines: Since the paper focused on PEFT fine-tuning with a novel architecture, there should be more comparison against PEFT techniques in the literature, such as co-training, probing (only fine-tung a final layer), LoRA fine-tuning [1], or model merging [2].
- Ablations: It is unclear how important the pre-backbone layers are. There should be an ablation of whether the APB outperforms only fine-tuning the post-backbone layers (a.k.a probing).
- OOD environments: since the paper focuses on performance improvement on OOD environments, it is important to define and explain in the main paper what these OOD environments are and how they are OOD.

[1] LoRA: Low-Rank Adaptation of Large Language Models
[2] Robust Finetuning of Vision-Language-Action Robot Policies via Parameter Merging

---

> ### Author Rebuttal · Authors · 2026-03-30
>
> We thank the reviewer for the feedback.
>
> **Answer to weakness 1 and question 2.**
> We agree that Algorithm 1 should have been explained more clearly. Although the manuscript already states that the actor and critic use standard SAC formulations, this was not tied clearly enough to Algorithm 1.
>
> APB decomposes the policy into a shared backbone and task-specific linear layers before and after it. The head and tail denote these task-specific layers, i.e., $\rho_i=\{\rho_i^h,\rho_i^t\}$ for task $T_i$. In the meta-trained setting, the shared backbone $\psi$ is learned across meta-training tasks, while each task has its own head/tail and critic (Q-function). During meta-training, for each task $T_i$, we collect transitions using $\pi(s;\rho_i,\psi)$, update the task-specific critic with $L_{\mathrm{critic}}^i$, update the task-specific linear layers with $L_{\mathrm{actor}}^i$, and then update $\psi$ using the average actor loss across tasks. During OOD adaptation, we load the frozen shared backbone $\psi$, initialize a new task-specific head/tail $\rho$ and critic $\omega$, and update only these task-specific components. In the randomly initialized backbone setting, $\psi$ is not meta-trained; it is randomly initialized and frozen, and only the task-specific head/tail and critic are learned.We will clarify these in the final version.
>
> **Answer to weakness 2 and weakness 3.**
> We agree that APB can also be viewed from a PEFT perspective. That said, the goal of this paper is not a broad benchmark of generic PEFT methods, but OOD adaptation in the meta-RL setting. We therefore center comparisons on standard meta-RL baselines from the same problem setting. Broader PEFT methods such as LoRA or model merging are interesting, but were developed in different regimes and are therefore less direct in our setting.
>
> We also agree that probing (i.e., tuning only the post-backbone layer) is the most direct baseline for APB and the key ablation for the pre-backbone layer. However, our choice to adapt both the pre- and post-backbone linear layers was not arbitrary. Section 4 theoretically motivates linear adaptation before and after a shared backbone, and prior work on surgical fine-tuning suggests that input-level shifts can favor tuning earlier layers rather than only the last layer [1]. In our setting, changing the task objective/reward can substantially alter the states visited by the policy, inducing a shift in the policy's input distribution. This motivates including a pre-backbone adaptation layer rather than restricting adaptation to probing-only updates. We agree that an explicit probing comparison would strengthen the empirical picture, and we will include it in the final version.
>
> [1] Y. Lee, A. S. Chen, F. Tajwar, A. Kumar, H. Yao, P. Liang, and C. Finn, Surgical Fine-Tuning Improves Adaptation to Distribution Shifts, ICLR 2023.
>
> **Answer to weakness 4 and question 1.**
> The OOD setting is already specified in the paper, with task descriptions in Appendix~B and referenced from the main text. In our experiments, OOD means that the meta-test task lies outside the task distribution seen during meta-training, e.g., adapting to negative target velocity after meta-training only on target velocities in $[0,3]$, as well as horizon extrapolation beyond the training range. We will clarify this in the final version.
>
> **Answer to question 3.**
> Figure~4 compares two adaptation strategies under the same network scale and SAC-based training framework: (i) APB with a randomly initialized frozen backbone and trainable pre-/post-backbone linear layers, and (ii) a standard RL policy with randomly initialized parameters trained end-to-end. Thus, the comparison is not simply between freezing and unfreezing the same APB model, but between APB's structurally constrained adaptation and standard full fine-tuning from random initialization.
>
> This is one of the most interesting findings of the paper. Section~4.2 shows that the theoretical analysis does not impose specific conditions on the backbone policy in its simplified setting, which is consistent with the empirical observation that even a randomly initialized frozen backbone can remain competitive when combined with task-specific linear adaptation. This suggests that the APB structure itself provides a strong inductive bias: by freezing the backbone and restricting adaptation to lightweight task-specific linear parameters, APB can avoid some instability of full end-to-end fine-tuning while remaining expressive enough for online adaptation.
>
> This experiment is evaluated under extrapolative conditions, since we increase episode length beyond training and force the policy into unseen states. This makes generalization more important than pure fitting to the training regime, and may help explain why a structurally constrained model with a frozen random backbone can remain competitive with, and in some cases outperform, the standard end-to-end baseline. We will clarify this in the final version.

---

> > ### Author Rebuttal · Reviewer_C71Z · 2026-04-01
> >
> > Thanks for the author's response. I have adjusted my score accordingly. While the choice of pre- and post- layers are motivated, it would still be useful to see empirical comparisons.

---

> > > ### Author Response · Authors · 2026-04-03
> > >
> > > Thank you for the positive update and for the helpful discussion. We will incorporate all the discussed points in the revised version.
> > >
> > > We additionally conducted probing under the exact same experimental setting as in Figure 4, where only the final layer was fine-tuned, and evaluated the final performance at the end of training. We will include the full results in the final version.
> > >
> > > | Method | cheetah vel | cheetah dir | ant dir | ant goal |
> > > |---|---:|---:|---:|---:|
> > > | Probing (only fine-tuning the final layer) | -212.30 $\pm$ 84.37 | 653.78 $\pm$ 110.09 | 185.18 $\pm$ 3.43 | -610.52 $\pm$ 16.13 |
> > > | Ours | -40.04 $\pm$ 2.69 | 1221.85 $\pm$ 25.46 | 259.36 $\pm$ 83.74 | -227.79 $\pm$ 35.52 |

---

### Official Review · Reviewer_nRdp · 2026-03-11

**Soundness:** 3
**Presentation:** 4
**Significance:** 4
**Originality:** 4
**Overall Recommendation:** 5
**Confidence:** 5

**Summary:**

The new method Adaptive Policy Backbone is introduced, which consists of splitting the deep neural network that is at the core of deep reinforcement learning tasks into a core backbone, and a linear input layer and a linear output layer. This split is shown to improve generalization to out of distribution tasks, and is shown to exhibit transfer learning. In doing so, it makes a major contribution to reinforcement  learning, which normally does not generalize to different tasks. There is a limited theoretical analysis, and there are experimental results on MuJoCo tasks. The method is not tried on pixel-based games (Atari). Also, the explanation why it works is somewhat shallow. The method, with parameter efficient finetuning, is somewhat inspired by the field of LLM. Nevertheless, the results are quite interesting, the experiments are elaborate, and this work should be published to the community.

**Compliance With Llm Reviewing Policy:**

Affirmed.

**Final Justification:**

My questions have been adequately addressed. I maintain my recommendation for the paper to be accepted.

**Key Questions For Authors:**

- Please provide a more detailed explanation concerning insights why this works
- Please increase font size of legends
- I would be very interested in results of runs on Atari, and explanations when it does not work.
- Are results correlated to the architecture of the inner backbone? Provide table with performance improvement versus network architecture.

**Limitations:**

Yes

**Strengths And Weaknesses:**

*Soundness*

The theoretical proof appears sound, and the experiments are described well, with enough detail to allow reproduction. The experimental validation is performed against all important other meta-RL methods.

*Presentation*

The paper is very well written, the english is good, and the work is very easy to follow. The details of theoretical proof and experimental evidence are clearly described. The legends of the figures are much too small, they are unreadable when the pdf is printed on paper. Please enlarge the font in the camera ready verison.

*Significance*

Generalization in RL is an important problem, and this work claims a major improvement. This is important work that appears sound, and should be published.

*Originality*

The work by D'Eramo uses a similar approach, this work is referenced correctly, and the difference is explained. The approach is original, and the results are surprising. This is an important result.

---

> ### Author Rebuttal · Authors · 2026-03-30
>
> We thank the reviewer for the positive assessment of the paper and for the constructive questions. Below, we address each question in turn and clarify the corresponding points.
>
> **Answer to question~1.**
> We agree that the paper should explain more clearly why APB works. Our current interpretation is that APB works because adaptation is restricted to linear transformations before and after a fixed shared backbone. In this structure, the backbone provides a reusable nonlinear projection across tasks, while the task-specific linear layers adapt how each task is mapped into and out of that shared representation. This is also consistent with our theoretical analysis, which motivates adaptation through linear transformations around a shared backbone in a simplified setting.
>
> Our experiments suggest that two effects contribute. First, the APB structure itself is important: even a randomly initialized frozen backbone can remain competitive, which indicates that the modular pre/post-linear adaptation already provides a strong inductive bias. Second, the quality of the shared representation still matters in harder regimes: in behavior cloning from restricted-support demonstrations, a meta-trained backbone shows a clearer advantage, suggesting that stronger priors become more important when adaptation must generalize from limited data. We will make this mechanism-level explanation clearer in the final version.
>
> **Answer to question~2.**
> Thank you for pointing this out. We agree that the legend font size should be increased for print readability, and we will enlarge the figure legends in the camera-ready version.
>
> **Answer to question~3.**
> We appreciate this suggestion. Pixel-based domains such as Atari introduce an additional layer of difficulty beyond the MuJoCo-style meta-RL benchmarks considered in our paper, since they require learning visual representations in addition to adaptation across tasks [1,2]. Classical few-shot meta-RL methods have also typically been evaluated on continuous-control benchmarks rather than Atari-like visual domains [3,4]. Moreover, recent work evaluating meta-RL on vision-based benchmarks including Atari reports that this setting poses substantially different challenges from those targeted in standard continuous-control meta-RL studies [2]. At the same time, we agree that extending APB to pixel-based environments is an interesting and important direction. As already discussed in the limitations, visual-domain evaluation such as Atari is beyond the scope of the current study and remains an important direction for future work.
>
> [1] T. Z. Zhao, A. Nagabandi, K. Rakelly, C. Finn, and S. Levine, MELD: Meta-Reinforcement Learning from Images via Latent State Models, CoRL 2021.
>
> [2] Z. Mandi, P. Abbeel, and S. James, On the Effectiveness of Fine-Tuning Versus Meta-Reinforcement Learning, NeurIPS 2022.
>
> [3] K. Rakelly, A. Zhou, D. Quillen, C. Finn, and S. Levine, Efficient Off-Policy Meta-Reinforcement Learning via Probabilistic Context Variables, ICML 2019.
>
> [4] L. M. Zintgraf, K. Shiarlis, V. Kurin, K. Hofmann, and S. Whiteson, Fast Context Adaptation via Meta-Learning, ICML 2019.
>
> **Answer to question~4.**
> We appreciate this valuable question. The current random-backbone experiments show that APB’s modular structure itself can already yield competitive adaptation performance, indicating that the performance gain is not solely tied to one particular learned backbone. However, we agree that these results do not fully reveal how the choice of backbone architecture (e.g., depth, width, or architectural family) affects the magnitude of improvement. For more complex tasks, the expressiveness of the policy backbone may become a limiting factor, and more expressive architectures such as transformer- or diffusion-based backbones could be a promising direction to explore. We therefore agree that a systematic comparison across backbone architectures would be valuable, and we will explicitly mention this as an important direction for future work.

---

> > ### Author Rebuttal · Reviewer_nRdp · 2026-04-03
> >
> > My questions have been adequately addressed. I maintain my recommendation for the paper to be accepted.

---

> > > ### Author Response · Authors · 2026-04-04
> > >
> > > We really appreciate the reviewer's time and effort in helping us strengthen this work.

---

### Official Review · Reviewer_L5rg · 2026-03-13

**Soundness:** 3
**Presentation:** 3
**Significance:** 3
**Originality:** 3
**Overall Recommendation:** 4
**Confidence:** 3

**Summary:**

The paper focuses on the problem that standard RL methods often suffer from poor generalization ability when encountering new tasks, especially under out-of-distribution (OOD) settings. To address this issue, the authors propose a simple yet effective framework called Adaptive Policy Backbone (APB), which consists of a shared frozen backbone and two task-specific linear layers placed before and after the backbone. The policy is first meta-trained across multiple tasks. During adaptation to new tasks, the backbone remains frozen while only the task-specific layers are updated. Experimental results show that the proposed method can effectively improve OOD generalization.

**Compliance With Llm Reviewing Policy:**

Affirmed.

**Final Justification:**

The paper proposes a network architecture for improving OOD generalization and provides detailed theoretical analysis. While the empirical evaluation is relatively limited and the practical impact remains somewhat unclear, the theoretical contribution is meaningful. Overall, I lean toward a weak accept, but I would not strongly object to rejection.

**Key Questions For Authors:**

Please refer to the Weaknesses section above.

**Limitations:**

yes

**Strengths And Weaknesses:**

Strengths:

- The paper presents clear and informative visualizations that help illustrate the proposed method and its behavior.

- The paper provides concrete theoretical analysis, which helps strengthen the intuition behind the proposed architecture.

- The paper includes comprehensive experimental results across multiple meta-RL benchmarks.

Weaknesses:

- The novelty of the proposed method is somewhat questionable. Previous works have already explored various similar architectural designs in meta-RL to improve generalization, such as different variants of Decision Transformers and other representation-sharing approaches.

- The application scenario of the method is somewhat unclear. The proposed approach appears similar to a meta-RL method, yet it significantly outperforms standard meta-RL baselines despite its relatively simple design. It would be helpful if the authors could clarify why this occurs and whether the method can be combined with existing meta-RL approaches.

- The assumption regarding OOD generalization seems somewhat narrow. Meta-RL typically addresses a broader range of task adaptation scenarios, and it is unclear what practical settings correspond to the specific OOD formulation considered in this paper.

---

> ### Author Rebuttal · Authors · 2026-03-30
>
> We thank the reviewer for the careful reading and constructive feedback. Below, we respond to the main concerns regarding novelty, application scenario, and the scope of the OOD formulation, and clarify how we will improve the corresponding discussion in the final version.
>
> **Answer to weakness~1.**
> We agree that the broad idea of combining shared and task-specific components is not new in reinforcement learning, and related directions have been explored in shared-representation multi-task RL [1] and shared-backbone architectures with task-specific layers [2]. However, our contribution is more specific than proposing another modular architecture. Prior work mainly studies how to combine shared and task-specific modules within a larger policy architecture, whereas our paper shows that effective task adaptation can be achieved by keeping the backbone fixed and adapting only linear layers before and after it. In other words, our claim is not merely that shared representations are useful, but that a much more constrained and interpretable adaptation rule is sufficient, as supported by both our simplified theoretical analysis and empirical OOD results.
>
> [1] C. D’Eramo, D. Tateo, A. Bonarini, M. Restelli, and J. Peters, Sharing Knowledge in Multi-Task Deep Reinforcement Learning, ICLR 2020.
>
> [2] J. S. Roberts and J. Di, Projected Task-Specific Layers for Multi-Task Reinforcement Learning, ICRA 2024.
>
> **Answer to weakness~2.**
> We agree that the presentation could be clearer on this point. APB is not intended as a universal replacement for meta-RL, but a structurally constrained framework for OOD adaptation where tasks vary mainly through reward/objective changes.
>
> APB can outperform standard meta-RL baselines despite its simple design because our paper focuses specifically on OOD task adaptation, which remains a difficult regime for many standard meta-RL methods. Existing baselines often rely either on broad parameter updates or on context/latent inference mechanisms trained on the meta-training distribution; both can become brittle when the test task induces trajectories outside that distribution. In contrast, APB imposes a stronger structural constraint: it freezes the shared backbone and adapts only lightweight pre-/post-backbone linear layers. We believe this is precisely why APB can outperform more flexible meta-RL baselines in the OOD setting---its simplicity acts as a strong inductive bias and reduces the risk of instability during adaptation.
>
> As discussed in Section 4.2 and validated in Figure 4, this advantage is not solely due to preserving a meta-trained backbone; even a randomly initialized frozen backbone can remain competitive, suggesting that APB's robustness arises largely from the architecture itself, while meta-trained representations help more in harder regimes such as behavior cloning from restricted data support.
>
> We also view APB as an adaptation parameterization rather than a mutually exclusive meta-RL paradigm. In the current paper, APB is instantiated with standard SAC-style actor-critic updates, so it can in principle be combined with alternative meta-training objectives or inference schemes, although such hybrids are beyond our scope.
>
> **Answer to weakness~3.**
> More broadly, task shift in meta-RL can arise through both reward variation and transition variation [1,2]. The latter has been especially prominent as an OOD generalization axis in sim2real, where a substantial literature studies robustness through dynamics/domain randomization by training over a broader distribution of environments [3,4]. In this paper, we instead focus on reward/objective-driven OOD adaptation under shared dynamics. This regime is also well established, but more commonly studied through transfer-RL formulations such as successor features and related reward-generalization methods [2]. It is also practically meaningful: even when the environment dynamics remain unchanged, a new objective can drive the policy toward state-action regions not emphasized during meta-training, as in deployment settings with unseen target velocities, directions, or goals, or under longer-horizon execution beyond the training range [5].
>
> [1] J. Beck, R. Vuorio, E. Z. Liu, Z. Xiong, L. Zintgraf, C. Finn, and S. Whiteson, A Tutorial on Meta-Reinforcement Learning, Foundations and Trends in Machine Learning 2025.
>
> [2] A. Barreto, W. Dabney, R. Munos, J. J. Hunt, T. Schaul, H. van Hasselt, and D. Silver, Successor Features for Transfer in Reinforcement Learning, NeurIPS 2017.
>
> [3] X. B. Peng, M. Andrychowicz, W. Zaremba, and P. Abbeel, Sim-to-Real Transfer of Robotic Control with Dynamics Randomization, ICRA 2018.
>
> [4] F. Muratore, F. Ramos, G. Turk, W. Yu, M. Gienger, and J. Peters, Robot Learning from Randomized Simulations: A Review, Frontiers in Robotics and AI 2021.
>
> [5] R. Poiani, A. Tirinzoni, and M. Restelli, Meta-Reinforcement Learning by Tracking Task Non-stationarity, IJCAI 2021.

---

> > ### Author Rebuttal · Reviewer_L5rg · 2026-04-04
> >
> > I thank the authors for addressing my concerns. I will maintain my evaluation.

---

> > > ### Author Response · Authors · 2026-04-04
> > >
> > > We really appreciate the reviewer's time and effort in helping us strengthen this work.

---

### Decision · Program_Chairs · 2026-04-30

**Decision:**

Accept (spotlight)

**Comment:**

The paper receives three weak accepts and one accept, indicating a clear positive consensus. Reviewers agree that the topic is important, the theoretical analysis is solid, and the experimental results are generally strong and convincing. Across the discussion, earlier concerns have been satisfactorily addressed in the rebuttal, with no remaining major objections.

Overall, the reviewers converge on a favorable evaluation, with only minor reservations remaining.